# Permutation Decision Trees using Structural Impurity

## Abstract

Decision Tree is a well understood Machine Learning model that is based on minimizing impurities in the internal nodes. The most common impurity measures are *Shannon entropy* and *Gini impurity*. These impurity measures are insensitive to the order of training data and hence the final tree obtained is invariant to a permutation of the data. This leads to a serious limitation in modeling data instances that have order dependencies. In this work, we use *Effort-To-Compress* (ETC) - a complexity measure, for the first time, as an impurity measure. Unlike Shannon entropy and Gini impurity, structural impurity based on ETC is able to capture order dependencies in the data, thus obtaining potentially different decision trees for different permutation of the same data instances (*Permutation Decision Trees*). We then introduce the notion of *Permutation Bagging* achieved using permutation decision trees without the need for random feature selection and sub-sampling. We compare the performance of the proposed permutation bagged decision trees with Random Forest. Our model does not assume independent and identical distribution of data instances. Potential applications include scenarios where a temporal order is present in the data instances.

## 1 Introduction

The assumptions in Machine Learning (ML) models play a crucial role in interpretability, reproducibility, and generalizability. One common assumption is that the dataset is independent and identically distributed (iid). However, in reality, this assumption may not always hold true, as human learning often involves connecting new information with what was previously observed. Psychological theories such as Primacy and Recency Effects [1], Serial Position Effect, and Frame Effect suggest that the order in which data is presented can impact decision-making processes. In this work, we have devised a learning algorithm that exhibits sensitivity to the order in which data is shuffled. This unique characteristic imparts our proposed model with decision boundaries or decision functions that rely on the specific arrangement of training data.

In our research, we introduce the novel use of 'Effort to Compress' (ETC) as an impurity function for Decision Trees, marking the first instance of its application in Machine Learning. ETC effectively measures the effort required for lossless compression of an object through a predetermined lossless compression algorithm [2]. ETC was initially introduced in [3] as a measure of complexity for timeseries analysis, aiming to overcome the limitations of entropy-based complexity measures. It is worth noting that the concept of complexity lacks a singular, universally accepted definition. In [2], complexity was explored from different perspectives, including the effort-to-describe (Shannon entropy, Lempel-Ziv complexity), effort-to-compress (ETC complexity), and degree-of-order (Subsymmetry). The same paper highlighted the superior performance of ETC in distinguishing between periodic and chaotic timeseries. Moreover, ETC has played a pivotal role in the development of an interventional causality testing method called Compression-Complexity-Causality (CCC) [4]. The effectivenss CCC has been tested in various causality discovery applications [5, 6, 7, 8]. ETC

has demonstrated good performance when applied to short and noisy time series data, leading to its utilization in diverse fields such as investigating cardiovascular dynamics [9], conducting cognitive research [10], and analysis of muscial compositions [11]. The same is not the case with entropy based methods.

In this research, we present a new application of ETC in the field of Machine Learning, offering a fresh perspective on its ability to capture structural impurity. Leveraging this insight, we introduce a decision tree classifier that maximizes the ETC gain. It is crucial to highlight that Shannon entropy and Gini impurity fall short in capturing structural impurity, resulting in an impurity measure that disregards the data's underlying structure (in terms of order). The utilization of ETC as an impurity measure provides the distinct advantage of generating different decision trees for various permutations of data instances. Consequently, this approach frees us from the need to adhere strictly to the i.i.d. assumption commonly employed in Machine Learning. Thus, by simply permuting data instances, we can develop a Permutation Decision Forest.

The paper is structured as follows: Section 2 introduces the Proposed Method, Section 3 presents the Experiments and Results, Section 4 discusses the Limitations of the research, and Section 5 provides the concluding remarks and outlines the future work.

## 2    Proposed Method

In this section, we establish the concept of structural impurity and subsequently present an illustrative example to aid in comprehending the functionality of ETC.

*Definition:* Structural impurity for a sequence $S = s_0, s_1, \ldots, s_n$, where $s_i \in \{0, 1, \ldots, K\}$, and $K \in \mathbf{Z}^+$ is the the extent of irregularity in the sequence $S$.

We will now illustrate how ETC serves as a measure of structural impurity. The formal definition of ETC is the effort required for lossless compression of an object using a predefined lossless compression algorithm. The specific algorithm employed to compute ETC is known as Non-sequential Recursive Pair Substitution (NSRPS). NSRPS was initially proposed by Ebeling [12] in 1980 and has since undergone improvements [13], ultimately proving to be an optimal choice [14]. Notably, NSRPS has been extensively utilized to estimate the entropy of written English [15]. The algorithm is briefly discussed below: Let's consider the sequence $S = 00011$ to demonstrate the iterative steps of the algorithm. In each iteration, we identify the pair of symbols with the highest frequency and replace all non-overlapping instances of that pair with a new symbol. In the case of sequence $S$, the pair with the maximum occurrence is $00$. We substitute all occurrences of $00$ with a new symbol, let's say 2, resulting in the transformed sequence $2011$. We continue applying the algorithm iteratively. The sequence $2011$ is further modified to become $311$, where the pair $20$ is replaced by 3. Then, the sequence $311$ is transformed into $41$ by replacing $31$ with 4. Finally, the sequence $41$ is substituted with 5. At this point, the algorithm terminates as the stopping criterion is achieved when the sequence becomes homogeneous. ETC, as defined in [3], represents the count of iterations needed for the NSRPS algorithm to attain a homogeneous sequence.

We consider the following three sequence and compute the ETC:

Table 1: Comparison of ETC with Shannon entropy, and Gini impurity for various binary sequences.

| Sequence ID | Sequence | ETC | Entropy | Gini Impurity |
|---|---|---|---|---|
| A | 111111 | 0 | 0 | 0 |
| B | 121212 | 1 | 1 | 0.5 |
| C | 222111 | 5 | 1 | 0.5 |
| D | 122112 | 4 | 1 | 0.5 |
| E | 211122 | 5 | 1 | 0.5 |

Referring to Table 1, we observe that for sequence A, the ETC, Shannon Entropy, and Gini impurity all have a value of zero. This outcome arises from the fact that the sequence is homogeneous, devoid of any impurity. Conversely, for sequences B, C, D, and E, the Shannon entropy and Gini impurity remain constant, while ETC varies based on the structural characteristics of each sequence. Having shown that the ETC captures the structural impurity of a sequence, we now define *ETC Gain*. ETC

gain is the reduction in ETC caused by partioning the data instances according to a particular attribute of the dataset. Consider the decision tree structure provided in Figure 1.

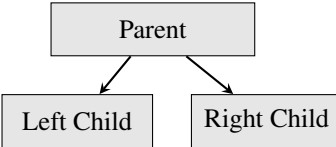

Figure 1: Decision Tree structure with a parent node and two child node (Left Child and Right Child).

The ETC Gain for the chosen parent attribute of the tree is defined as follows:

$$ETC\_Gain = ETC(Parent) - [w_{Left\_Child} \cdot ETC(Left\_Child) + w_{Right\_Child} \cdot ETC(Right\_Child)], \tag{1}$$

where $w_{Left\_Child}$ and $w_{Right\_Child}$ are the weights associated to left child and right child respectively. The formula for ETC Gain, as given in equation 1, bears resemblance to information gain. The key distinction lies in the use of ETC instead of Shannon entropy in the calculation. We now provide the different steps in the *Permutation Decision Tree* algorithm.

1. Step 1: Choose an attribute to be the root node and create branches corresponding to each possible value of the attribute.

2. Step 2: Evaluate the quality of the split using ETC gain.

3. Step 3: Repeat Step 1 and Step 2 for all other attributes, recording the quality of split based on ETC gain.

4. Step 4: Select the partial tree with the highest ETC gain as a measure of quality.

5. Step 5: Iterate Steps 1 to 4 for each child node of the selected partial tree.

6. Step 6: If all instances at a node share the same classification (homogeneous class), stop developing that part of the tree.

# 3   Experiments and Results

To showcase the effectiveness of the ETC impurity measure in capturing the underlying structural dependencies within the data and subsequently generating distinct decision trees for different permutations of input data, we utilize the following illustrative toy example.

Table 2: Toy example dataset to showcase the potential of a permuted decision tree generated with a novel impurity measure known as "Effort-To-Compress".

| Serial No. | $f_1$ | $f_2$ | label |
|---|---|---|---|
| 1 | 1 | 1 | 2 |
| 2 | 1 | 2 | 2 |
| 3 | 1 | 3 | 2 |
| 4 | 2 | 1 | 2 |
| 5 | 2 | 2 | 2 |
| 6 | 2 | 3 | 2 |
| 7 | 4 | 1 | 2 |
| 8 | 4 | 2 | 2 |
| 9 | 4 | 3 | 1 |
| 10 | 4 | 4 | 1 |
| 11 | 5 | 1 | 1 |
| 12 | 5 | 2 | 1 |
| 13 | 5 | 3 | 1 |
| 14 | 5 | 4 | 1 |

The visual representation of the toy example provided in Table 2 is represented in Figure 2

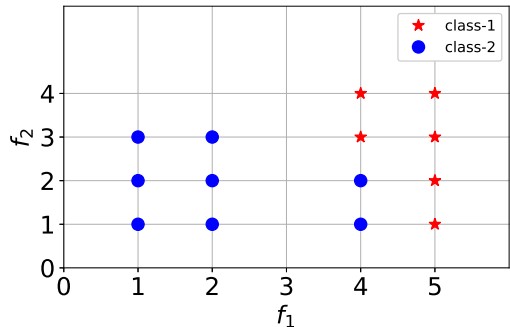

Figure 2: A visual representation of the toy example provided in Table 2.

We consider the following permtation of dataset, for each of the below permutation we get distinct decision tree.

- Serial No. Permutation A: 1, 2, 3, 4, 5, 6, 7, 8, 9, 10, 11, 12, 13, 14. Figure 3 represents the corresponding decision tree.

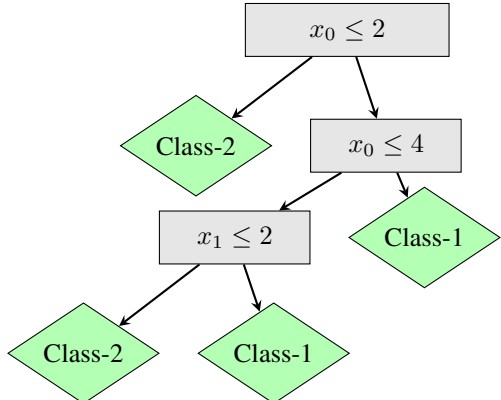

Figure 3: Decision using ETC for Serial No. Permutation A.

- Serial No Permutation B: 14, 3, 10, 12, 2, 4, 5, 11, 9, 8, 7, 1, 6, 13. Figure 4 represents the corresponding decision tree.

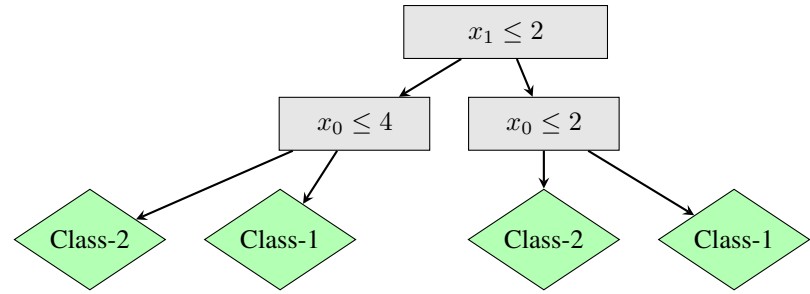

Figure 4: Decision Tree using ETC for Serial No. Permutation B.

- Serial No Permutation C: 13, 11, 8, 12, 7, 6, 4, 14, 10, 5, 2, 3, 1, 9. Figure 5 represents the corresponding decision tree.

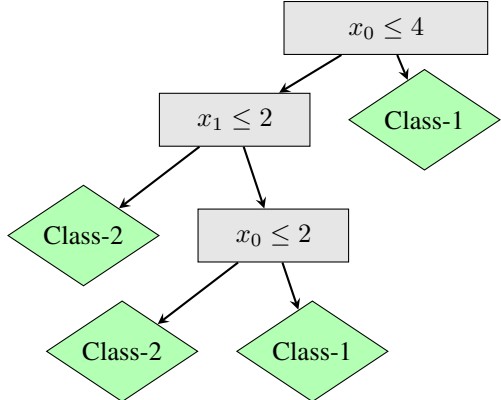

Figure 5: Decision Tree using ETC for Serial No. Permutation C.

- Serial No Permutation D: 3, 2, 13, 10, 11, 1, 4, 7, 6, 9, 8, 14, 5, 12. Figure 6 represents the corresponding decision tree.

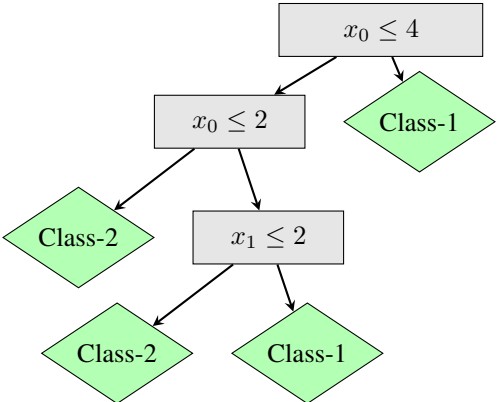

Figure 6: Decision Tree using ETC for Serial No. Permutation D.

- Serial No Permutation E: 10, 12, 1, 2, 13, 14, 8, 11, 4, 7, 9, 6, 5, 3. Figure 7 represents the corresponding decision tree.

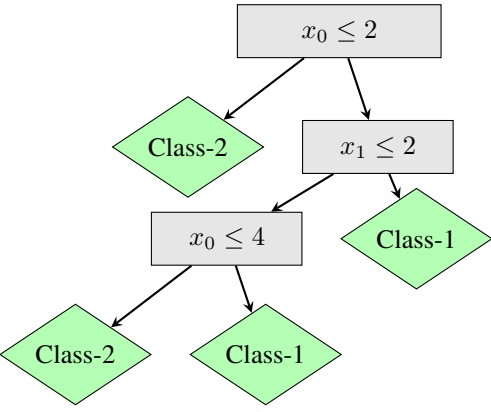

Figure 7: Decision Tree using ETC for Serial No. Permutation E.

The variability in decision trees obtained from different permutations of data instances (Figures 3, 4, 5, 6,and 7) can be attributed to the ETC impurity function's ability to capture the structural impurity of labels, which sets it apart from Shannon entropy and Gini impurity. Table 3 highlights the sensitivity of ETC to permutation, contrasting with the insensitivity of Shannon entropy and Gini impurity towards data instance permutations. In the given toy example, there are six class-1 data instances and eight class-2 data instances. Since Shannon entropy and Gini impurity are probability-based methods, they remain invariant to label permutation. This sensitivity of ETC to the structural pattern of the label motivates us to develop a bagging algorithm namely Permutation Decision Forest.

Table 3: Comparison between Shannon Entropy, Gini Impurity and Effort to Compress for the toy example.

| Label Impurity | Shannon Entropy (bits) | Gini Impurity | Effort-To-Compress |
|---|---|---|---|
| Permutation A | 0.985 | 0.490 | 7 |
| Permutation B | 0.985 | 0.490 | 8 |
| Permutation C | 0.985 | 0.490 | 9 |
| Permutation D | 0.985 | 0.490 | 9 |
| Permutation E | 0.985 | 0.490 | 8 |

## 3.1 Permutation Decision Forest

Permutation decision forest distinguishes itself from Random Forest by eliminating the need for random subsampling of data and feature selection in order to generate distinct decision trees. Instead, permutation decision forest achieves tree diversity through permutation of the data instances. The accompanying architecture diagram provided in Figure 8 illustrates the operational flow of permutation decision forest.

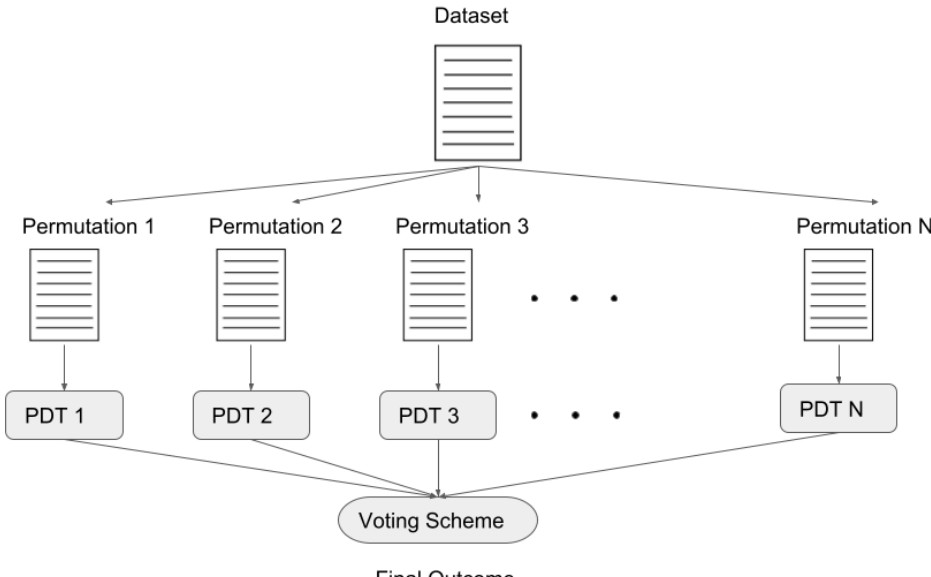

Figure 8: Architecture diagram of Permutation Decision Forest. Permutation Decision Forest, which comprises multiple individual permutation decision trees. The results from each permutation decision tree are then fed into a voting scheme to determine the final predicted label.

The architecture diagram depicted in Figure 8 showcases the workflow of the Permutation Decision Forest, illustrating its functioning. Consisting of individual permutation decision trees, each tree operates on a permuted dataset to construct a classification model, collectively forming a strong

classifier. The outcomes of the permutation decision trees are then fed into a voting scheme, where the final predicted label is determined by majority votes. Notably, the key distinction between the Permutation Decision Forest and Random Forest lies in their approaches to obtaining distinct decision trees. While Random Forest relies on random subsampling and feature selection, Permutation Decision Forest achieves diversity through permutation of the input data. This distinction is significant as random feature selection in Random Forest may result in information loss, which is avoided in Permutation Decision Forest.

## 3.2   Performance comparison between Random Forest and Permutation Decision Forest

We evaluate the performance of the proposed method with the following datasets: *Iris* [16], *Breast Cancer Wisconsin* [17], *Haberman's Survival* [18], *Ionosphere* [19], *Seeds* [20], *Wine* [21]. For all datasets, we allocate 80% of the data for training and reserve the remaining 20% for testing. Table 4 provides a comparison of the hyperparameters used and the test data performance as measured by macro F1-score.

Table 4: Performance comparison of Permutation Decision Forest with Random Forest for various publicly available datasets

| Dataset | Random Forest | | | Permutation Decision Forest | | |
|---|---|---|---|---|---|---|
| | F1-score | n_estimators | max_depth | F1-score | n_estimators | max_depth |
| Iris | 1.000 | 100 | 3 | 0.931 | 31 | 10 |
| Breast Cancer Wisconsin | 0.918 | 1000 | 9 | 0.893 | 5 | 10 |
| Haberman's Survival | 0.560 | 1 | 3 | 0.621 | 5 | 10 |
| Ionosphere | 0.980 | 1000 | 4 | 0.910 | 5 | 5 |
| Seeds | 0.877 | 100 | 5 | 0.877 | 11 | 10 |
| Wine | 0.960 | 10 | 4 | 0.943 | 5 | 10 |

In our experimental evaluations, we observed that the proposed method surpasses Random Forest (F1-score = 0.56) solely for the Haberman's survival dataset (F1-score = 0.621). However, for the Seeds dataset, the permutation decision forest yields comparable performance to Random Forest (F1-score = 0.877). In the remaining cases, Random Forest outperforms the proposed method.

## 4   Limitations

The current framework demonstrates that the proposed method, permutation decision forest, achieves slightly lower classification scores compared to random forest. We acknowledge this limitation and aim to address it in our future work by conducting thorough testing on diverse publicly available datasets. It is important to note that permutation decision trees offer an advantage when dealing with datasets that possess a temporal order in the generation of data instances. In such scenarios, permutation decision trees can effectively capture the specific temporal ordering within the dataset. However, this use case has not been showcased in our present work. In our future endeavors, we intend to incorporate and explore this aspect more comprehensively.

## 5   Conclusion

In this research, we present a unique approach that unveils the interpretation of the *Effort-to-Compress* (ETC) complexity measure as an impurity measure capable of capturing structural impurity in timeseries data. Building upon this insight, we incorporate ETC into Decision Trees, resulting in the introduction of the innovative *Permutation Decision Tree*. By leveraging permutation techniques, Permutation Decision Tree facilitates the generation of distinct decision trees for varying permutations of data instances. Inspired by this, we further develop a bagging method known as *Permutation Decision Forest*, which harnesses the power of permutation decision trees. Moving forward, we are committed to subjecting our proposed method to rigorous testing using diverse publicly available datasets. Additionally, we envision the application of our method in detecting adversarial attacks.

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
