# OpenReview forum: "Permutation Decision Trees using Structural Impurity"
_NeurIPS.cc/2023/Conference — Submitted to NeurIPS 2023_

### Official Review · Reviewer_7dHM · 2023-06-24

**Soundness:** 1 poor
**Presentation:** 2 fair
**Contribution:** 1 poor
**Rating:** 2
**Confidence:** 4

**Summary:**

The authors proposed the permutation decision trees method, which uses Effort-To-Compress as the impurity measure to model the order dependencies of data instances, and extended the proposed permutation decision tree to a variant of random forest. They also did some experiments to compare the performance of the proposed methods with random forests.

**Strengths:**

The proposed structural impurity can actually capture the order dependencies of data instances, as shown in the examples in Table 1.

**Weaknesses:**

The paper exhibits several weaknesses, which are outlined below:

1. Insufficient clarity regarding the chosen setting: The authors' intended focus appears to be on time series data; however, the task discussed pertains to multi-class classification, which is an i.i.d. setting. The authors are recommended to formalize the problem setup.
2. Inconsistent use of notation: The paper demonstrates inconsistencies in notation usage. For instance, the features presented in Table 3 are denoted as $f_{1}, f_{2}$, whereas in Figures 3-7, they are represented as $x_{0}, x_{1}$.
3. Unfair experimental setup and insignificant results: The experiment setup lacks fairness, and the obtained results do not exhibit statistical significance. In the only out-performing dataset, the random forest model employed only one tree, while the proposed permutation decision forest utilized five trees, indicating an apparent unfairness in the comparison. Furthermore, the hyperparameters' n_estimators vary across different datasets, which is deemed unreasonable.

**Questions:**

1. Which type of tasks does this paper care about? See Weaknesses #1.
2. If the concerned tasks are order-sensitive, why can the data instances be casually permuted when generating a forest?

**Limitations:**

The authors adequately addressed the limitations in Section 4.

---

> ### Author Rebuttal · Authors · 2023-08-06
>
> **Addressing Weakness 1**: The applicability of the proposed method is not limited to just timeseries data.  In our paper, we are interested in the use case where order in which data instances are presented plays a crucial role, we are not interested in the dependency on a single data instance. In the modified manuscript, we have added the following section titled “Temporal vs. Spatial Ordering in Decision Making” to address the comments.
>
> In the toy example (Table 2), the data instances were represented spatially (refer to Figure 2). However, it is also possible to imagine the data instances to be events in time. This has significant bearing on the decision making process.
>
> Imagine that there is an outbreak of a new virus (like COVID-19) and data instances (rows of Table 2) correspond to chronological admission of patients to a testing facility. The decision to be taken is to quarantine (if virus is present) or not quarantine the patient (if no virus present). Let $f_0$ represent the severity (including diversity) of symptoms of the incoming patient and $f_1$ represent the value obtained by a molecular test performed on a bio-sample (for eg., blood sample) taken from the patient. The labels $1$ and $2$ correspond to the presence of virus (POSITIVE) and absence of the virus (NEGATIVE) respectively. Consequently, the patient with POSITIVE label is quarantined. It is also the case that determining the severity (and diversity) of symptoms is relatively straightforward as it involves a thorough examination by the attending physician. On the other hand, performing the molecular test on the patient's blood sample involves considerable cost and time, and not to mention the discomfort the patient is subjected to. The decision making process needs to factor in all these additional constraints.
>
> Now, the permutations A-E of the data correspond to different realities in which the events happen over time. Even though all the $14$ patients with exactly the same severity of symptoms and molecular test value occur (we assume that the molecular test is done on all the patients for the purposes of medical records), the order in which they arrive is different in each of these realities. Consequently, the decision tree obtained by our method is going to be different in each reality which is intuitive and reasonable. Conventional DT with information gain or Gini impurity would make no distinction between these realities and yield the same DT in each reality. However, this would not be ideal as we shall show.
>
> To illustrate the contrasting decision trees obtained by the ETC based impurity measure, consider the trees obtained for permutation B and permutation D. We have re-drawn them below to better aid the comparison (refer to Figure 8 and Figure 9). Both decision trees fit the same data (up to a permutation) and hence have identical performance metrics. However, the former needs molecular testing of all $14$ patients whereas the later needs the testing to be done on only $4$ patients. The argument to be made here is that in the reality encountered by permutation D, the incoming patients arrive in a particular order where it becomes necessary to decide to perform the molecular testing on all of them. Whereas, in the alternate reality corresponding to permutation B, the patients arrive in such an order that the decision to test first on severity of symptoms is the correct one. This has a huge impact on future events -- in Figure 8, each and every patient in the future will also be subjected to the molecular testing whereas not in the case of Figure 9. Thus, decision making strictly depends on the chronological ordering of events - as is the case in real life.
>
> Please find the link to content we have added in the updated manuscript, Figures for Permutation B and D are provided in the below link: https://drive.google.com/file/d/1o91TpYjnbj-OZ7fHUeuSLhKnRwVSxPyA/view?usp=sharing
>
> **Addressing Weakness 2**: Thank you for pointing it out. In the revised manuscript, we have corrected the same.
>
> **Addressing Weakness 3**:  In response to the reviewer's feedback, we have made significant improvements to the manuscript. We have now included a dedicated section that presents a thorough performance comparison between the permutation decision tree and the classical decision tree, using various real-world datasets.
> To ensure a fair and robust evaluation, we have provided detailed information on the hyperparameter tuning process. We conducted cross-validation experiments to validate the effectiveness of both approaches, and the results have been included in the revised manuscript.
> Furthermore, we have included the test results obtained from the comparison, providing comprehensive insights into the performance of the permutation decision tree in comparison to the classical decision tree. Please find the screenshot of the results and insights:https://drive.google.com/file/d/1RzuO_-3Hyo96vKQKxMQLtVcVekw9UDzv/view?usp=sharing
> In the revised manuscript, we have added the hyperparameter tuning details of Random Forest.
>
> **Response to Questions**: In this paper, our focus lies on tasks involving data instances with temporal dependencies. Although the dataset need not be limited solely to time series data, we are interested in scenarios where each data instance exhibits a dependency on its preceding data instance. Our interest extends to encompass various cases where the relationships between consecutive data instances play a crucial role in the underlying patterns and decision-making process. These dependencies could manifest in diverse domains, and our proposed method aims to effectively capture and leverage them for enhanced performance and interpretability.

---

> > ### Comment · Reviewer_7dHM · 2023-08-17
> >
> > Thank you for providing such a comprehensive explanation. However, I regret to mention that the issues I raised remain unresolved.
> >
> > I'm struggling to grasp the precise scope of the settings the authors intend to address, as it appears to be a somewhat ambiguously defined problem. Regrettably, the revised version of the paper still lacks a concrete mathematical formulation of the dependency in question. In light of this, I would like to suggest that the authors focus on clarifying the following points in their forthcoming iteration:
> >
> > 1. What is dependency in mathematics? (for example, a transition rule in time series analysis)
> > 2. What is the objective in the concerned setting? (for example, minimizing the accumulated loss in times series analysis)
> >
> > I believe that addressing these aspects in the next version of the paper will substantially enhance its comprehensibility and value.

---

> > > ### Author Response · Authors · 2023-08-20
> > > **Thank you**
> > >
> > > Thank you for the valuable comments. We will work on the suggestion in the next version.

---

### Official Review · Reviewer_maQx · 2023-06-26

**Soundness:** 2 fair
**Presentation:** 3 good
**Contribution:** 2 fair
**Rating:** 3
**Confidence:** 4

**Summary:**

In traditional decision tree algorithms such as CART and C4.5, impurity measures are used to split internal nodes. The paper proposes a decision tree induction method by using effort-to-compress (ETC) measure, which can capture order dependencies in the data. With ETC’s ability to capture order dependencies, permuting the data can result in different trees, thereby constructing a forest without the need for bagging. This proposed decision tree induction method can be used for datasets with temporal order.

**Strengths:**

The paper uses ETC as a new impurity measure for constructing decision trees. Since ETC is sensitive to the order of data points, the tree built using this measure may be well-suited for temporal datasets. And it also provides a different way for constructing diverse trees and thereby getting a forest. Overall, the paper is clearly written and easy to follow.

**Weaknesses:**

- What about the bias and variance in the permutation decision forest? Random forest uses bagging and random feature selection to make trees in the forest uncorrelated, thereby reducing variance. But trees in the permutation forest are not uncorrelated. Using the ensemble of these correlated trees may not reduce variance.

- In the toy example, some leaf paths are shown in different trees. I am wondering if there will be a significant number of duplicated leaf paths within the permutation forest.

- Experiments only show the comparison between random forest and permutation tree forest in terms of F1-score. How about other evaluation metrics, e.g. misclassification loss? The results don’t show that the proposed method outperforms random forest. And there is no comparison between the performance of single trees, such as CART vs. ETC tree.


**Questions:**

I am wondering how the hyperparameters are chosen in the experiments. Random forest tends to have lower depth but more estimators, while permutation tree forest has deeper trees with fewer estimators.

**Limitations:**

The authors have addressed the limitations of the paper and propose future directions.

---

> ### Author Rebuttal · Authors · 2023-08-06
>
> **Addressing Weakness 1**: In the revised manuscript, we have thoughtfully included a dedicated section titled "Model vs. Domain Interpretability, Temporal Generalizability, and Causal Decision Learning." In this section, we have explored the interpretability aspects of our proposed model, highlighting the key differences between our permutation decision tree and the random forest.: The procedure of bagging that is employed in Random forests is problematic from the perspective of interpreatability or explainability. Leaving out some of the features and a random sampling of data instances is bound to result in biased decision trees. Even though the final classification performance of Random forest may be good, the use of biased trees due to random sampling and feature subsampling leads to a loss of interpretability and reliability in the decision making process. It should be noted that leaving out a particular feature or a data instance for constructing a decision tree (as is the case in Random forest algorithm) is completely unjustified from the point of view of the application domain. It is an arbitrary adhoc step that has no valid justification from an explainability/interpretability point of view. Consider the scenario where the left out data instance is an anomaly or a rare event with physical or engineering significance. For example, the data instance that has been removed pertains to a system overload event or a fault event in the monitoring of a power system. In a cyber security scenario, the left out data instance could be an adversarial attack (which is sparse and very rare event).  The objective or the goal of the learning task in these applications is in fact to model such rare/extreme/anomalous events in order to understand and garner insights, in which case, the decision tree used to arrive at the final classification rule in Random forest is completely non-intuitive.
>
> In Permutation Decision forest, every permutation of the data instance corresponds to an `alternate' reality (a counterfactual?!) where that particular order of the data instances are presented to the algorithm to result in a specific set of decisions made subsequently by the classifier. Two different permutations of the same data could mean/signify entirely two different realities or states of the world. If each data instance corresponds to a specific time, then a different ordering corresponds to a different temporal sequence of events - which is clearly a very different reality to one that produce the training data.  Random forest has no way of capturing these counterfactual realities. Thanks to the sensitivity of the structural impurity measure to data ordering, Permutation Decision forest is able to efficiently capture this via different decision making rules. In effect, what Permutation Decision forest is learning is a {\it generalized} set of decision rules that are invariant (under all permutations) to temporal re-ordering. This is a form of generalization which is missing in Random forest. We could call this type of generalization -- a form of \emph{temporal generalization} that respects counterfactual realities.   Thus, Permutation Decision forest is a pre-cursor to a causality informed decision tree algorithm. Future research work will focus on making  these causal underpinnings more explicit and pronounced with suitable enhancements to upgrade Permutation Decision forest to a full-blow causal reasoning/ causal decision learning algorithm.
>
> **Addressing Weakness 2**: It is possible that different permutation may sometimes give the same tree (if the ETC values are similar).
>
> **Addressing Weakness 3**:  In response to the reviewer's feedback, we have made significant improvements to the manuscript. We have now included a dedicated section that presents a thorough performance comparison between the permutation decision tree and the classical decision tree, using various real-world datasets.
> To ensure a fair and robust evaluation, we have provided detailed information on the hyperparameter tuning process. We conducted cross-validation experiments to validate the effectiveness of both approaches, and the results have been included in the revised manuscript.
> Furthermore, we have included the test results obtained from the comparison, providing comprehensive insights into the performance of the permutation decision tree in comparison to the classical decision tree. Please find the screenshot of the results and insights:https://drive.google.com/file/d/1RzuO_-3Hyo96vKQKxMQLtVcVekw9UDzv/view?usp=sharing
> In the revised manuscript, we have added the hyperparameter tuning details of Random Forest.
>
> **Response to Question**: Please go through weakness 3, where we describe how the hyperparameter tuning is done.

---

> > ### Comment · Reviewer_maQx · 2023-08-16
> >
> > I want to thank the authors for their detailed rebuttal. However, I believe the proposed methodology still needs more work. I will maintain my current score.
> >
> > Looking at Table 4, DT outperforms PDT on some datasets. This makes me doubt if PDT is the best option.
> >
> > I appreciate adding a discussion about interpretability and temporal generalizability. I agree that a single PDT with fewer leaves is interpretable. But when there are many individual trees in the permutation decision forest, there could be many decision rules and these rules may not always be consistent. How can you ensure interpretability when it becomes a tree ensemble?
> >
> > The new section also mentions model interpretability and domain interpretability. Showing multiple DTs to domain experts and allowing them to select the most meaningful decision tree relate to the idea of Rashomon set or model multiplicity. But it is not clear to me how PDT or permutation decision forest outperforms a single optimal decision tree and the Rashomon set of trees.

---

> > > ### Author Response · Authors · 2023-08-20
> > > **Thank you**
> > >
> > > Thank you for the valuable comments.  When it comes to tree ensembles, the Permutation Decision Forest distinguishes itself by abstaining from any form of random sub-sampling or feature selection. In contrast to conventional methods, the Permutation Decision Forest preserves the entirety of its features, utilizing each one in its decision-making process. This approach results in the generation of distinct decision trees through the permutation of data instances alone. This commitment to utilizing all features is pivotal for achieving interpretability, as it ensures that no potentially crucial attributes are overlooked. Unlike the utilization of random features, which could inadvertently omit vital information, the Permutation Decision Forest effectively mitigates this concern.

---

### Official Review · Reviewer_Q9hV · 2023-07-05

**Soundness:** 1 poor
**Presentation:** 1 poor
**Contribution:** 1 poor
**Rating:** 2
**Confidence:** 5

**Summary:**

The paper "Permutation Decision Trees using Structural Impurity" introduces a novel split criterion for the training of decision trees that also takes the order of labels inside the training data into account. This way, to obtain a forest, one only needs to shuffle the data before training individual trees. Moreover, the novel split criterion supposedly works better for data that includes (temporal) dependencies, although der are no experiments to support this claim.

**Strengths:**

- I think the idea of tackling non-iid data with a novel split criterion is nice, and Effort-To-Compress as an impurity measure seems like a good choice

**Weaknesses:**

- The experimental evaluation is very weak. The authors compare their method on 6 small real-world datasets and one artificial dataset and compare it only against Random Forests. Moreover, their method seems to be worse compared to RF. Hyperparameters are also incomparable, as the RF uses smaller trees than their method although it is well-known that RF benefits more from larger trees. In addition, the number of estimators changes for every experiment. There is no clear experimental protocol, and the authors do not use random repetitions and/or cross-validation but resort to a single train/test split. The experimental evaluation is hence borderline useless and can only be seen as a first test-experiment.
- The paper contains limited valuable information. While the Effort-To-Compress (ETC) measure seems to be of central interest here, the authors do not present a formal mathematical explanation of it. They mention the NSRPS algorithm to compute ETC, but also do not explain it mathematically, and only offer a single example. A thorough mathematical explanation and the typical explanations of the notation (Model function f(x), samples X, labels Y, etc.) is missing entirely.
- The authors deal with the case in which the order of samples is important. This is completely against the typical IID assumption we have in Machine Learning. Unfortunately, the authors neither discuss this (certainly interstring) difference in more detail nor do they really present any real-world example of this.
- A dedicated Related Work section is missing, although there is plenty of space left in the paper. The authors decided to waste roughly two pages by printing different DTs, which does not add any new information to the paper. This space would have been used better to highlight related work or pinpoint the novelty of this work in more detail.
- Eq. (1) and Tab. 4 do not fit the page width

**Questions:**

- Are there any datasets and/or real-world applications you are aware of in which the order of samples matter? What are the implications for "classical" IID ML here?

**Limitations:**

The authors acknowledge that their method is worse compared to the state of the art and intend to perform more testing. As this paper presents a novel method I don't see any immediate negative societal impact.

---

> ### Author Rebuttal · Authors · 2023-08-06
>
> **Addressing Weakness 1**:  In response to the reviewer's feedback, we have made significant improvements to the manuscript. We have now included a dedicated section that presents a thorough performance comparison between the permutation decision tree and the classical decision tree, using various real-world datasets.
> To ensure a fair and robust evaluation, we have provided detailed information on the hyperparameter tuning process. We conducted cross-validation experiments to validate the effectiveness of both approaches, and the results have been included in the revised manuscript.
> Furthermore, we have included the test results obtained from the comparison, providing comprehensive insights into the performance of the permutation decision tree in comparison to the classical decision tree. Please find the screenshot of the results and insights:https://drive.google.com/file/d/1RzuO_-3Hyo96vKQKxMQLtVcVekw9UDzv/view?usp=sharing
> In the revised manuscript, we have added the hyperparameter tuning details of Random Forest.
>
> **Addressing Weakness 2**: In the supplementary materials, we have provided the mathematical explanation of NSRPS  algorithm.
>
> **Addressing Weakness 3**:  In our paper, we are interested in the use case where order in which data instances are presented plays a crucial role, we are not interested in the dependency on a single data instance. In the modified manuscript, we have added the following section titled “Temporal vs. Spatial Ordering in Decision Making” to address the comments.
>
> In the toy example (Table 2), the data instances were represented spatially (refer to Figure 2). However, it is also possible to imagine the data instances to be events in time. This has significant bearing on the decision making process.
>
> Imagine that there is an outbreak of a new virus (like COVID-19) and data instances (rows of Table 2) correspond to chronological admission of patients to a testing facility. The decision to be taken is to quarantine (if virus is present) or not quarantine the patient (if no virus present). Let $f_0$ represent the severity (including diversity) of symptoms of the incoming patient and $f_1$ represent the value obtained by a molecular test performed on a bio-sample (for eg., blood sample) taken from the patient. The labels $1$ and $2$ correspond to the presence of virus (POSITIVE) and absence of the virus (NEGATIVE) respectively. Consequently, the patient with POSITIVE label is quarantined. It is also the case that determining the severity (and diversity) of symptoms is relatively straightforward as it involves a thorough examination by the attending physician. On the other hand, performing the molecular test on the patient's blood sample involves considerable cost and time, and not to mention the discomfort the patient is subjected to. The decision making process needs to factor in all these additional constraints.
>
> Now, the permutations A-E of the data correspond to different realities in which the events happen over time. Even though all the $14$ patients with exactly the same severity of symptoms and molecular test value occur (we assume that the molecular test is done on all the patients for the purposes of medical records), the order in which they arrive is different in each of these realities. Consequently, the decision tree obtained by our method is going to be different in each reality which is intuitive and reasonable. Conventional DT with information gain or Gini impurity would make no distinction between these realities and yield the same DT in each reality. However, this would not be ideal as we shall show.
>
> To illustrate the contrasting decision trees obtained by the ETC based impurity measure, consider the trees obtained for permutation B and permutation D. We have re-drawn them below to better aid the comparison (refer to Figure 8 and Figure 9). Both decision trees fit the same data (up to a permutation) and hence have identical performance metrics. However, the former needs molecular testing of all $14$ patients whereas the later needs the testing to be done on only $4$ patients. The argument to be made here is that in the reality encountered by permutation D, the incoming patients arrive in a particular order where it becomes necessary to decide to perform the molecular testing on all of them. Whereas, in the alternate reality corresponding to permutation B, the patients arrive in such an order that the decision to test first on severity of symptoms is the correct one. This has a huge impact on future events -- in Figure 8, each and every patient in the future will also be subjected to the molecular testing whereas not in the case of Figure 9. Thus, decision making strictly depends on the chronological ordering of events - as is the case in real life.
>
> Please find the link to content we have added in the updated manuscript, Figures for Permutation B and D are provided in the below link: https://drive.google.com/file/d/1o91TpYjnbj-OZ7fHUeuSLhKnRwVSxPyA/view?usp=sharing
>
>
> **Addressing Weakness 4 and 5**:In the revised manuscript, we have added relevant previous works. We have also modified Equation 1 and Table 4.
> **Response to Question**: Please go through the response to Weakness 3. We have provided an example where the use case of the method is highlighted

---

> > ### Comment · Reviewer_Q9hV · 2023-08-15
> >
> > After reading the author's response, I am still not convinced.
> >
> > **Addressing Weakness 1**: I acknowledge the added comparison of PDT vs DT in the provided screenshot. However, the results you are now showing are inconsistence. For example, on the Iris dataset looking at the cross-validated results, PDT beats DT, yet they have the same performance on the test set. On the breast cancer dataset, the effect is even worse: DT beats PDT when looking at cross-validation but loses on the test set. I understand that cross-validation is difficult when the order of data is essential (and there are a few papers discussing this in the time-series domain, although I am unsure how applicable they are. See e.g. [1]), but since the values are partially contradicting each other, I am still unsure which method is the best.
> >
> > **Addressing Weakness 2**: Unfortunately, I cannot find the supplementary material for this paper. Is this part of the rebuttal?
> >
> > **Addressing Weakness 3**: I am sorry for the miscommunication here. I appreciate the additional example, but I still feel there is a lack of notation and proper mathematical explanation. In the typical Machine Learning setting, we assume that data instances are i.i.d samples. Now the authors introduce an algorithm that also respects the order of samples in the dataset. Hence the underlying assumption is that certain permutations might occur more frequently and/or certain permutations impact certain labels (as shown in the example). Now I wonder: How can we formally, i.e. in a mathematical sense, characterize this dependency? I believe this is the core issue I have with the paper, as a proper mathematical model would also help with the proper evaluation of the model.
> >
> > [1] "A note on the validity of cross-validation for evaluating autoregressive time series prediction" by Bergmeir et al. 2018

---

### Official Review · Reviewer_iLmx · 2023-07-07

**Soundness:** 2 fair
**Presentation:** 2 fair
**Contribution:** 4 excellent
**Rating:** 4
**Confidence:** 5

**Summary:**

The paper proposes a novel in Decision Tree literature splitting criteria based on Effort To Compress (ETC) gain. Use of this criteria is justified by a desire to work with data that doesn't conform to i.i.d. assumption about the generating distribution. There is an experiment on a synthetic data that shows that different decision trees are generated when different orderings of the data are used for training. A permutation voting forest is introduced, that allows using random permutations of full data to obtain multiple different decision trees for use in the final ensemble of trees. There is an evaluation of Permutation Voting Forest against regular random forests on multiple real world datasets that however show slightly lower results when using proposed method.

**Strengths:**

- The paper opens a novel line of research about using Decision Trees for modeling data, that comes in a sequence and does not follow i.i.d. assumption.
- There is a novel application of ETC measure as a splitting criteria in decision trees.
- A generalization of the proposed model: Permutation Decision Forest is introduced, that uses a novel idea of shuffling the data in the context of a splitting criteria that generates different trees for different permutations of the training data.


**Weaknesses:**

- It is noted that usage of ETC allows to get rid of i.i.d. assumption. But this claim needs more thorough theoretical analysis. If we want to keep sequential structure of the data, the sequence still gets destroyed upon split: split does not split examples in a consecutive way; some examples may go to the left split, then some to the right, then again to the left part of the split and so on. So, new left and right sequences after the split will have completely different properties. Considering an example from introduction, where ETC is used: musical compositions. Splitting the musical composition according to some feature, like presence of some range of frequencies at a given moment, will result in an unpleasant music on both sides of the split, because instead of hearing half of the musical composition, we will hear a "fractal" - small parts of original sequence with small gaps inbetween, that got assigned to left or right parts of the split
- Related to the previous point: at the testing phase there is no "memory" in the model, and the model still predicts elements by looking at them one by one. So, shuffling the testing set will result in exactly same predictions. Can we say that the problem of non-i.i.d. distribution is solved, if the behavior on the testing set is equal to the behaviour of the i.i.d. models?
- Testing of regular Decision Trees with proposed splitting criteria on real data is needed (only the forests were tested on real data, but proposed forests work differently due to the proposed shuffling of the input data, so regular trees must be evaluated separately as well). It would be nice to both test on regular datasets (that are not sequentially ordered, like the datasets from section 3.2), and also to find at least some example real datasets where ordering is important, and where proposed model (regular permutation decision tree) is both practically and theoretically better than the baseline decision tree models.
- In section 3.2 a more thorough experimental design would be more convincing. (a) If we compare proposed model to the baseline, why hyperparameters are different for same dataset? If hyperparameters tuning was done, that should be thoroughly described. (b) Experiments should be run several times on different train-test splits and mean scores and standard deviations should be reported to allow fair comparison in the presense of noise.


**Questions:**

- The proposed decision tree and forest work very differently, because forest does internal data shuffling, which destroys ordering of the original input data. Shall we consider two variants of the forest: one with shuffling, and one without (where ETC splitting is used, but instead of shuffling we do regular feature and example subsets selection)? Then the forest with shuffling may be compared to original Random Forest (because both are not using sequential information); and the Forest without shuffling may be compared to the single proposed permutation decision tree.

- Minor note: typo on line 41: "muscial"


**Limitations:**

Limitations are well described in the paper, which is good. Since there were identified significant limitations in terms of accuracy of the proposed permutation decision forests (that may also affect proposed single permutation decision trees), it would be more convincing to include additional experiments that will clarify the extent of such limitations right away without deferring them to the future work.

---

> ### Author Rebuttal · Authors · 2023-08-06
>
> Dear reviewer,
>
> Thank you for the valuable feedback. Please find the point by point response to each of the weakness pointed out.
>
> **Addressing Weakness No 1**: In our paper, we are interested in the use case where order in which data instances are presented plays a crucial role, we are not interested in the dependency on a single data instance. In the modified manuscript, we have added the following section titled “Temporal vs. Spatial Ordering in Decision Making” to address the comments.
>
> In the toy example (Table 2), the data instances were represented spatially (refer to Figure 2). However, it is also possible to imagine the data instances to be events in time. This has significant bearing on the decision making process.
>
> Imagine that there is an outbreak of a new virus (like COVID-19) and data instances (rows of Table 2) correspond to chronological admission of patients to a testing facility. The decision to be taken is to quarantine (if virus is present) or not quarantine the patient (if no virus present). Let $f_0$ represent the severity (including diversity) of symptoms of the incoming patient and $f_1$ represent the value obtained by a molecular test performed on a bio-sample (for eg., blood sample) taken from the patient. The labels $1$ and $2$ correspond to the presence of virus (POSITIVE) and absence of the virus (NEGATIVE) respectively. Consequently, the patient with POSITIVE label is quarantined. It is also the case that determining the severity (and diversity) of symptoms is relatively straightforward as it involves a thorough examination by the attending physician. On the other hand, performing the molecular test on the patient's blood sample involves considerable cost and time, and not to mention the discomfort the patient is subjected to. The decision making process needs to factor in all these additional constraints.
>
> Now, the permutations A-E of the data correspond to different realities in which the events happen over time. Even though all the $14$ patients with exactly the same severity of symptoms and molecular test value occur (we assume that the molecular test is done on all the patients for the purposes of medical records), the order in which they arrive is different in each of these realities. Consequently, the decision tree obtained by our method is going to be different in each reality which is intuitive and reasonable. Conventional DT with information gain or Gini impurity would make no distinction between these realities and yield the same DT in each reality. However, this would not be ideal as we shall show.
>
> To illustrate the contrasting decision trees obtained by the ETC based impurity measure, consider the trees obtained for permutation B and permutation D. We have re-drawn them below to better aid the comparison (refer to Figure 8 and Figure 9). Both decision trees fit the same data (up to a permutation) and hence have identical performance metrics. However, the former needs molecular testing of all $14$ patients whereas the later needs the testing to be done on only $4$ patients. The argument to be made here is that in the reality encountered by permutation D, the incoming patients arrive in a particular order where it becomes necessary to decide to perform the molecular testing on all of them. Whereas, in the alternate reality corresponding to permutation B, the patients arrive in such an order that the decision to test first on severity of symptoms is the correct one. This has a huge impact on future events -- in Figure 8, each and every patient in the future will also be subjected to the molecular testing whereas not in the case of Figure 9. Thus, decision making strictly depends on the chronological ordering of events - as is the case in real life.
>
> Please find the link to content we have added in the updated manuscript, Figures for Permutation B and D are provided in the below link: https://drive.google.com/file/d/1o91TpYjnbj-OZ7fHUeuSLhKnRwVSxPyA/view?usp=sharing
>
> **Addressing Weakness 2**: As you correctly pointed out, in the current scenario, the shuffling of test data does not have an effect. However, this issue can be addressed by implementing a method where each test data instance is predicted individually. After predicting a test data instance, we include it in the training set and build a new model. We then utilize the updated model to predict the subsequent test data instance, and this process continues iteratively. By adopting this approach, we effectively incorporate memory into the model, enabling it to consider the evolving knowledge from previously predicted test data instances.
>
>
> **Addressing Weakness 3 and 4**: In response to the reviewer's feedback, we have made significant improvements to the manuscript. We have now included a dedicated section that presents a thorough performance comparison between the permutation decision tree and the classical decision tree, using various real-world datasets.
> To ensure a fair and robust evaluation, we have provided detailed information on the hyperparameter tuning process. We conducted cross-validation experiments to validate the effectiveness of both approaches, and the results have been included in the revised manuscript.
> Furthermore, we have included the test results obtained from the comparison, providing comprehensive insights into the performance of the permutation decision tree in comparison to the classical decision tree. Please find the screenshot of the results and insights:https://drive.google.com/file/d/1RzuO_-3Hyo96vKQKxMQLtVcVekw9UDzv/view?usp=sharing
> In the revised manuscript, we have added the hyperparameter tuning details of Random Forest.
> We believe that these additions significantly enhance the quality of the paper, addressing the concerns raised by the reviewer regarding the performance comparison. We are grateful for the valuable feedback, which has allowed us to strengthen our research and present more comprehensive and conclusive findings.

---

> > ### Comment · Reviewer_iLmx · 2023-08-18
> >
> > Thank you for detailed response. I acknowledge your clarifications, and I would like to consider the following, as of now not changing my evaluation of the paper:
> >
> > - On Weakness 1, you have provided an example, where f_1 is more "costly" to obtain than f_0. However in the real world (in other dataset or example, that consists of the same feature values and examples ordering, but with different meaning) it could be all the way around, where f_0 would be more costly. So, in my opinion, this example does not answer the questions in general, and provides only one specific case which is hard to generalize into a theory.
> > - On Weakness 2 I acknowledge your suggestion.
> > - Regarding your response about Weaknesses 3 and 4 I would like to note that a hyperparameters search over depths from 1 to 150 with step size 10 is not sufficient, because depths over 10 are very unlikely to be optimal, because they result in overfitting on such small datasets.  This corresponds to the resulting fact that only depths 1 and 11 were selected by hyperparameters search, and both are likely to be suboptimal, because range from 2 to 10 was not covered by the search. Since the hyperparameters space was not fully explored, the conclusions may not be valid in the possible presence of more optimal results.

---

### Author Rebuttal · Authors · 2023-08-06

Dear reviewer,

Thank you for the valuable feedback. We have  modified the manuscript and addressed the comments raised by the reviewers. Following are the changes made:

1. A concrete example that illustrates the practical use case of the proposed Permutation Decision Tree. Please find the link for the same: https://drive.google.com/file/d/1o91TpYjnbj-OZ7fHUeuSLhKnRwVSxPyA/view?usp=sharing

2. We have thoughtfully included a dedicated section titled "Model vs. Domain Interpretability, Temporal Generalizability, and Causal Decision Learning (Section 5)." In this section, we have explored the interpretability aspects of our proposed model, highlighting the key differences between our permutation decision tree and the random forest. Due to lack of space, please find the link to the contents of this section: https://drive.google.com/file/d/1-bn79_d9VYPLZ1QT3xQTNnBzp0UoVMTo/view?usp=sharing

3. In response to the reviewer's feedback, we have made significant improvements to the manuscript. We have now included a dedicated section that presents a thorough performance comparison between the permutation decision tree and the classical decision tree, using various real-world datasets.
To ensure a fair and robust evaluation, we have provided detailed information on the hyperparameter tuning process. We conducted cross-validation experiments to validate the effectiveness of both approaches, and the results have been included in the revised manuscript. Furthermore, we have included the test results obtained from the comparison, providing comprehensive insights into the performance of the permutation decision tree in comparison to the classical decision tree. Please find the screenshot of the results and insights:https://drive.google.com/file/d/1RzuO_-3Hyo96vKQKxMQLtVcVekw9UDzv/view?usp=sharing. In the revised manuscript, we have added the hyperparameter tuning details of Random Forest

4. The mathematical details of the Non-sequential Recursive Pair Substitution (NSRPS) algorithm has been added in the supplementary material.

5. Correction to the notational inconsistency pointed out by the reviewer.

6. We shall provide the GitHub link to the codes used to build the proposed model.

Thank you for the valuable feedback. The feedback helped us to do more experiments and further provide more insights about the proposed permutation decision tree.

---

### Decision · Program_Chairs · 2023-09-21

**Decision:**

Reject

**Comment:**

The paper introduces a new splitting criterion, based on the Effort-To-Compress complexity measure, to train decision trees in the context of order-dependent data.

The reviewers found that the idea is novel and certainly worth exploring. But the considered setting is not sufficiently well defined theoretically and the experimental validation is too limited.

The paper can not be accepted at this stage.